

# Risk assessment of *FLT3* and *PAX5* variants in B-acute lymphoblastic leukemia: a case–control study in a Pakistani cohort

Ammara Khalid[1], Sara Aslam[1], Mehboob Ahmed[1], Shahida Hasnain[1] and Aimen Aslam[2]

[1] Department of Microbiology & Molecular Genetics, Quaid-e-Azam Campus, University of the Punjab, Lahore, Pakistan
[2] Department of Statistics and Actuarial Science, Quaid-e-Azam Campus, University of the Punjab, Lahore, Pakistan

Corresponding author
Ammara Khalid,
amara.mmg@pu.edu.pk

## ABSTRACT

**AIMS**. B-cell acute lymphoblastic leukemia (B-ALL) is amongst the most prevalent cancers of children in Pakistan. Genetic variations in *FLT3* are associated with auto-phosphorylation of kinase domain that leads to increased proliferation of blast cells. Paired box family of transcription factor (*PAX5*) plays a critical role in commitment and differentiation of B-cells. Variations in *PAX5* are associated with the risk of B-ALL. We aimed to analyze the association of *FLT3* and *PAX5* polymorphisms with B cell leukemia in Pakistani cohort.

**METHODS**. We collected 155 B-ALL subject and 155 control blood samples. For analysis, genotyping was done by tetra ARMS-PCR. SPSS was used to check the association of demographic factors of SNPs present in the population with the risk of B-ALL.

**RESULTS**. Risk allele frequency A at locus 13q12.2 (rs35958982, *FLT3*) was conspicuous and showed positive association (OR = 2.30, CI [1.20–4.50], $P = 0.005$) but genotype frequency (OR = 3.67, CI [0.75–18.10], $P = 0.088$) failed to show any association with the disease. At locus 9p13.2 (rs3780135, *PAX5*), the risk allele frequency was significantly higher in B-ALL subjects than ancestral allele frequency (OR = 2.17, CI [1.37–3.43], $P = 0.000$). Genotype frequency analysis of rs3780135 polymorphism exhibited the protective effect (OR = 0.55, CI [0.72–1.83], $P = 0.029$). At locus 13q12.2 (rs12430881, *FLT3*), the minor allele frequency G (OR = 1.15, CI [1.37–3.43], $P = 0.043$) and genotype frequency (OR = 2.52, $P = 0.006$) reached significance as showed $p < 0.05$.

**CONCLUSION**. In the present study, a strong risk of B-cell acute lymphoblastic leukemia was associated with rs35958982 and rs12430881 polymorphisms. However, rs3780135 polymorphism showed the protective effect. Additionally, other demographic factors like family history, smoking and consanguinity were also found to be important in risk assessment. We anticipate that the information from genetic variations in this study can aid in therapeutic approach in the future.

## INTRODUCTION

According to the Punjab cancer registry report, acute lymphoblastic leukemia (ALL) is a predominant malignancy among children and it makes up most prevalent cancer in Punjab, Pakistan. The worldwide incidence rate is 1–4.75 per 100,000 people. In Pakistan ALL contributes to 17.9% of all cancers. It is characterized by mutation in blast cells in hematopoietic stem cells, spleen, neurons, gonads, lymph nodes, and hepatic cells (*Portell, Wenzell & Advani, 2013*). Although, B-ALL is very common in children but it may also occur in the adult populace (*Forero, Hernández & Rivas, 2013*). Several demographic parameters like gender, age, family history and biological factors also play an important role in the prevalence of disease. Other factors like exposure to UV, radiations, lifestyle may also act as risk factors (*Levine et al., 2016*; *Acharya et al., 2018*). Mutation in certain genes involved in different processes like apoptosis, proliferation and differentiation of B-cells may also cause B-ALL. These genetic alterations largely affect the prediction and therapeutic approach used for medication and therapy of ALL (*Tasian & Hunger, 2017*).

FMS-like tyrosine kinase (*FLT3*) belongs to class III receptor tyrosine kinase (RTK) family. Structurally, *FLT3* consists of an extracellular domain at the amino terminus. This domain comprises of immunoglobulin-like transmembrane region and intracellular juxta-membrane domain (JMD). At the carboxyl terminus, there are two kinase domains, separated by a kinase insert region (*Gilliland & Griffin, 2002*). *FLT3* is expressed in normal human bone marrow especially in CD34+ hematopoietic stem, brain (*Çakmak Görür et al., 2019*) and gonads (*Matthews et al., 1991*; *Small et al., 1994*) and encodes 1,000 amino acid protein in humans. In the hematopoietic tissues, binding of FL with its receptor causes auto-phosphorylation of tyrosine residues present in the kinase domain and stimulates growth of progenitor cells in the marrow and blood (*Marhäll et al., 2018*). This results in downstream activation of signaling pathways that are involved in regulation of cell cycle or apoptosis, including (PI3K), caspase-9 and Ras/Raf pathways and causes multiplied proliferation of cells, reduced cell apoptosis, and inhibition of B-cell differentiation (*Zhang & Broxmeyer, 2000*).

In hematologic malignancy, 70% to 100% increased expression of *FLT3* in acute myeloid leukemia (AML) and acute lymphoblastic leukemia (ALL) is reported previously (*Brown et al., 2005*; *Griffith et al., 2016*). Rosnet and colleagues reported that three out of five ALL subjects with increased expression of *FLT3* in leukemia blasts (*Rosnet et al., 1996*). Another study showed that up regulation of *FLT3* gene is a potential risk factor of leukemia (*Cheng et al., 2018*).

B-cell-specific activator protein (*PAX5*) encodes transcription factors that are the member of a paired box domain. *PAX5* plays imperative role in the commitment of B-cell lineage from blast cells as it controls the differentiation of a pro-B cell to pre-B cells (*Fuxa & Skok, 2007*; *Lang et al., 2007*). In pre-pro-B cells the immunoglobin gene rearrangement starts and matures into pro-B cells. Expression of PAX5 gene initiates from pro-B stage and terminates at pre-B stage. In late B-lymphoposis, PAX5 maintains the function of mature B-cells (*Shahjahani et al., 2015*).

In B-cell malignancies, *PAX5* act as an oncogene. Down-regulation of *PAX5* halts B-cells and reverts B-cell precursors (BCPs) to progenitors (pro B-cell stage) (*Schebesta et al., 2007*; *Carotta & Nutt, 2008*). Conversely, uncontrolled proliferation of the B-cells leads to the abnormal expression of *PAX5* in precursor cells and inhibit T-cell proliferation (*Souabni, Jochum & Busslinger, 2007*). It is reported that in childhood ALL, translocations and mutation in *PAX5* are more prevalent (*Bousquet et al., 2007*; *Nebral et al., 2009*; *Santoro et al., 2009*; *Iacobucci & Mullighan, 2017*). Alternative splicing of *PAX5* in exon 7 to exon 9 results into five isoforms. These isoforms are more expressed in primary B-cell lymphoma tissues and cancerous cell lines (*Zwollo et al., 1997*; *Arseneau et al., 2009*).

Previous studies showed that the presence of single nucleotide polymorphisms (SNPs) in genome maybe risk causing or protective for the disease and it may also alters the pharmacokinetic and pharmacodynamics properties of drugs (*Kumanayake, 2013*; *Pui, 2015*; *Tasian & Hunger, 2017*). We selected two non-synonomous SNPs including 13q12.2 (rs35958982, *FLT3*), 557 (Val > Ile) at position Chr13:28034336 (GRCh38.p12) and 9p13.2 (rs3780135, *PAX5*), 293 (Thr > Ile) at position Chr9:36840626 (GRCh38.p12). A synonomous SNP 13q12.2 (rs12430881, *FLT3*), (A > G) at position Chr13:28020665 (GRCh38.p12) was also selected. The change in amino acid sequence due to non-synonomous SNP alters the protein structure implicating its expression and function. Current study is designed to evaluate the role of *FLT3* and *PAX5* genes in B-cell lymphoblastic leukemia. For this purpose, a case control analysis was conducted to evaluate the polymorphic association of rs35958982, rs3780135 and rs12430881 with B-cell acute lymphoblastic leukemia (B-ALL) incidence.

## MATERIALS & METHODS

### Study subjects

The present study was conducted at the University of Punjab, Pakistan and granted ethical approval to carry out the study within its facilities (Ethical Application Ref: sbs/222/18). Blood samples were collected during the period of January 2017 to February 2017 from Children's Hospital, Lahore, Pakistan. Study population comprised of 155 cases and 155 controls younger than 15 years of age. The diagnostic criteria for B-ALL cases include B-cell positive markers (CD19, CD10, CD22, and CD20) confirmed by flow cytometry analysis. Cases with relapsed and newly diagnosed B-ALL were also included. All 310 subjects recruited were consented to participate in this study after filling the questionnaire. The subjects with any other type of leukemia, blood infectious disease, and B-ALL subjects older than 15 years of age were excluded from the study. Family history with cancer, parental consanguinity (first and second degree relatives) and smoking status (>100 cigarettes in lifetime) were gathered by questionnaire interviewed.

### Genotyping

Venous blood samples of cases and controls were collected in EDTA vials. DNA extraction was done using Sam brook 2001 organic protocol. The genes and SNPs associated with B-ALL were screened using DisGeNET platform (*Queralt-Rosinach et al., 2016*) and were verified by dbSNP database (*Sherry et al., 2001*). Presence of the selected SNPs in Pakistani

**Table 1  Tetra-ARMS primers.**

| Primer | Sequence (5′–3′) | Tm (°C) |
|---|---|---|
|  | TGTGACAAATTAGCAGGGTTAACAC | 57.3 |
| rs35958982 | CACAGAAGAGATCACAGAAGGAGTCT | 60.7 |
|  | GAAACTCCCATTTGAGATCATATTCA | 56.0 |
|  | AGACAGAGACAAGCAGACATTCG | 58.4 |
|  | CTCTTCCAGGCTCCCCCGAC | 59.2 |
| rs3780135 | GGGCGGCAGCGCTATAAGAA | 59.5 |
|  | ACCCCAGCTCTAGATGGCGAAG | 56.6 |
|  | ATAGGTGCCATCAGTGTTTGGTGC | 58.4 |
|  | GTTTGTCTCCTCTTCATTGGCA | 56.0 |
| rs12430881 | GCCTCAGTGTCATCTTCGAATT | 56.3 |
|  | CCTTTTATCTTCACATCAGGCCT | 56.6 |
|  | CTTAGTAGAGATGGGGTTTTGCC | 58.4 |

**Table 2  PCR program for SNPs.**

| PCR steps | Temperature (°C) | Duration of steps | No. of cycles |
|---|---|---|---|
| Initial duration | 92 | 5 min | |
| Denaturation | 94 | 30 sec | |
| Annealing rs35958982 | 58.4 | | |
| rs3780135 | 56.6 | 1 min | 30–35 |
| rs12430881 | 58.8 | | |
| Extension | 72 | 1 min | |
| Final extension | 72 | 5 min | |

population was confirmed by Ensembl genome browser (*Frankish et al., 2017*). In order to identify the SNPs, tetra arm primers were designed using Primer1 software (*Ye et al., 2001*) as shown in Table 1. Tetra arms PCR was done using advanced primus 96 (PeqLab) thermal cycler (Table 2). PCR products were further analyzed by gel electrophoresis (Figs. 1–3).

## Statistical analysis

Statistical studies were performed using IBM SPSS 23. Chi-square test was conducted to compare categorical data. Allele and genotype association between SNPs and B-ALL were calculated by computing odds ratio (OR) and 95% confidence interval (CI). The Bonferroni corrections were applied for all multiple tests. A logistic regression model was used to adjust different B-ALL risk factors. The probability level accepted for significance was $P < 0.05$.

## RESULTS

Family history of cancer and parental consanguinity showed significant association with B-ALL while, there was no association with the smoker parents. Subjects with a family history of any type of cancer showed a high risk of having B-ALL (OR = 15.42, $P = 0.000$).
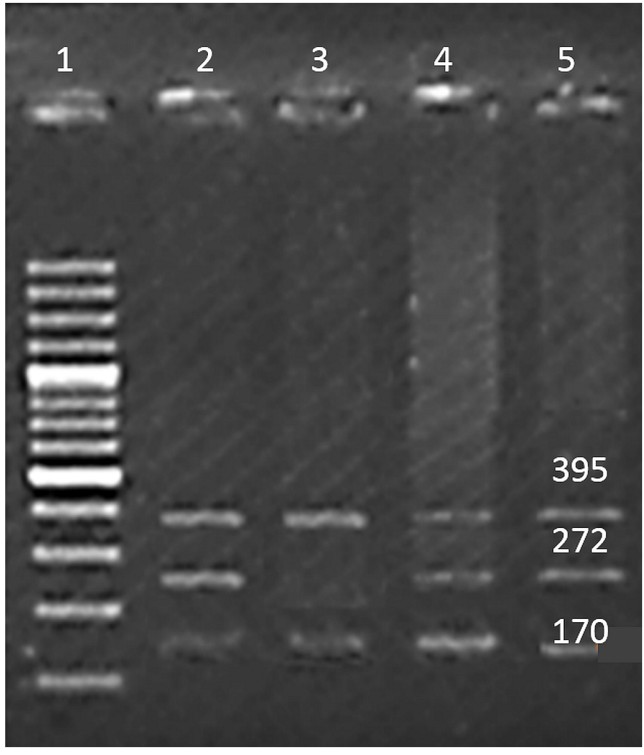

**Figure 1   SNP rs35958982.** Well 1 indicates the DNA ladder (100 bp), an amplicon (395 bp) is outer band. Amplicon 272 bp: allele 'A' and amplicon of 170 bp allele 'G'.

Previous studies showed smoking as a risk factor for cancer but our cohort displayed a contradictory results as no significant association was found in B-ALL subjects (OR = 0.85, $P = 0.580$). In the present study, more B-ALL subjects were product of parental consanguinity and showed highly significant association with the risk of B-ALL (OR = 1.87, $P = 0.050$) (Table 3).

Our data showed that none of the subjects and their parents was exposed to radiations. Furthermore, 18 patients had liver hepatomegaly sized 11.7 ± 3.3 mm, nine cases had nephropathy of right kidney 9.47 ± 2.9 mm and left kidney 10.01 ± 2.3 mm. Symptoms like night sweating, dizziness, abdominal pain, vomiting, bruises, pallor, enlarged lymph nodes, cough with blood, loose stools, jaundice, pedal edema, pain, dehydration, hepatosplenomegaly, atypical blast cells mild abdominal ascites, low leukocytes and thrombocytopenia were also recorded.

In our cohort, rs35958982 encoding isloleucine form of codon frequency in subjects was 13.4% and 5.6% in controls. Moreover, statistical analysis showed positive association of allele frequency (OR = 2.30, CI [1.20–4.50], $P = 0.005$) and no association of genotype frequency (OR = 3.67, CI [0.75–18.10], $P = 0.088$) with the disease. Another polymorphism rs3780135, a minor allele frequency in subjects was 47.1% and 32.58% in controls showing positive association with B-ALL (OR = 2.17, CI [1.37–3.43], $P = 0.000$). The genotype frequency showed protective effect with B-ALL (OR = 0.55, CI [0.72–1.83], $P = 0.029$).

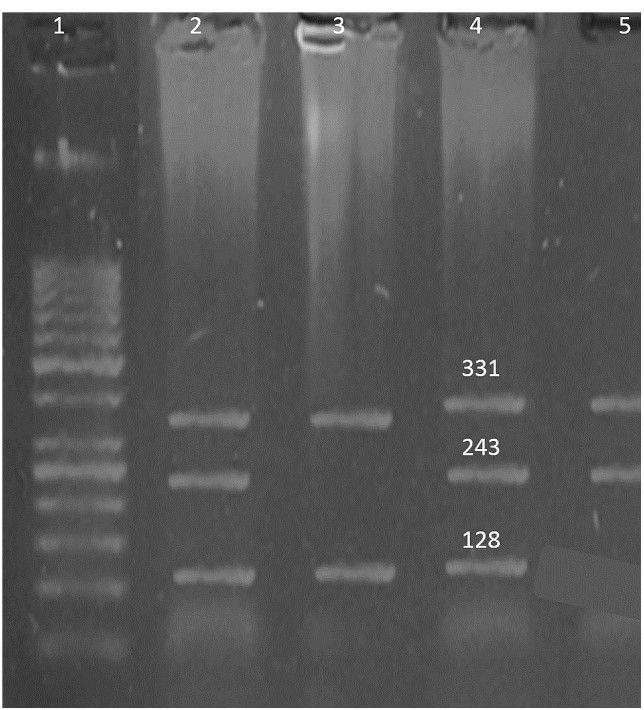

**Figure 2 SNP rs3780135.** Well 1 indicates the DNA ladder (50 bp), an amplicon (331 bp) is outer band. Amplicon 243 bp: allele 'A' and amplicon of 128 bp: allele 'G'.

The SNP rs12430881 allele frequency (OR = 1.15, CI [1.37–3.43], $P = 0.043$) and genotype frequency (OR = 2.52, CI [1.28–4.95], $P = 0.006$) showed strong association with the disease as shown in Table 4. After applying Bonferroni correction, SNPs rs35958982, rs3780135 and rs12430881 remained statistically significant and showed $P$-value 0.030, 0.010 and 0.002 respectively.

Multivariate regression analysis was performed after adjusting the baseline for conventional B-ALL risk factors such as family history, smoking and parental consanguinity. As shown in Table 4, the multivariate analysis indicated that outcome of heterozygous genotype GA in SNPs rs35958982, rs3780135 and rs12430881 had significant association with B-ALL and showed odds ratio (OR = 1.13, CI [0.41–3.08]), (OR = 1.19, CI [0.52–2.73]) and (OR = 1.09, CI [0.48–2.69]) respectively. Additionally, risk genotypes in SNPs rs35958982 (AA) and rs12430881 (GG) showed positive association with disease having an odds ratio (OD = 1.30, CI [0.37–5.08]) and (OD = 1.03, CI [0.50–2.68]), respectively. However, the risk genotype (AA) of SNP rs3780135 displayed no association with B-ALL after adjusting for environmental factors. Stratification analysis of environmental factors showed smoking as major risk factor in both heterozygous and risk genotype of SNP rs3780135 and rs12430881 whereas, parental consanguinity act as risk factor only in heterozygous genotype of SNP rs3780135 as shown in Table 5.
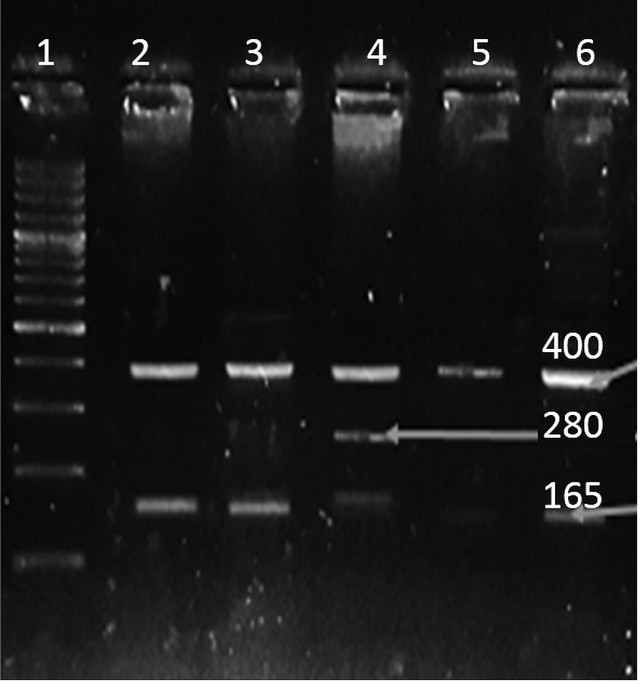

**Figure 3  SNP rs12430881.** Well 1 indicates the DNA ladder (100 bp), an amplicon (400 bp) is outer band. Amplicon 165 bp: allele G and amplicon of 280 bp: allele A.

**Table 3  Association of demographic factors with risk of B-ALL.**

| Parameters | Patients (%) | Control (%) | Odd ratio | Chi-square | P value |
|---|---|---|---|---|---|
| Age (mean) | 7.30 | 11.70 | | | |
| A positive family history | 16.77 | 1.29 | 15.42 | 14.59 | 0.000[*] |
| A negative family history | 83.22 | 98.70 | | | |
| Smoking by parent | 38.06 | 41.94 | 0.85 | 0.31 | 0.580 |
| No smoking parent | 61.93 | 58.06 | | | |
| Parental cousin marriage | 33.55 | 21.29 | 1.87 | 3.78 | 0.050[*] |
| No cousin marriage | 66.45 | 78.70 | | | |
| Females | 56 | 72 | 0.65 | 3.41 | 0.070 |
| Males | 99 | 83 | | | |

**Notes.**

Significant values are shown in (*).

# DISCUSSION

According to previous studies, association of first and second degree family history of cancer signifies genetic and environmental risk factor for causing acute lymphoblastic leukemia. Our study also showed positive association of family history with B-ALL (OR = 15.42, $P = 0.000$). Earlier, parental smoking has also been associated with the prevalence of ALL but our study showed contrary results (OR = 0.85, $P = 0.580$) (*Belson, Kingsley & Holmes, 2007*). Parental consanguinity is still practiced in Pakistan, which results in minor allele pool

**Table 4 Allele and genotype frequency.** Adjusted ORs were obtained from logistic regression model with adjustment for family history, smoking and consanguinity.

| Gene/SNP | Allele/ Genotype | Controls (%) | Cases (%) | Crude OR (95% CI) | Adjusted OR (95% CI) | $\chi^2$ | [a]P-value | [b]P-value |
|---|---|---|---|---|---|---|---|---|
| | Allele | | | | | | | |
| | A | 5.60 | 13.40 | 2.30 (1.20-4.50) | – | 7.79 | 0.005* | 0.002* |
| | G | 94.40 | 86.60 | | – | | | |
| rs35958982 | Genotype | | | | | | | |
| | GG | 90.70 | 79.60 | | 1.00 | | | |
| | GA | 7.60 | 13.80 | 3.67 (0.75-18.10) | 1.13 (0.41–3.08) | 2.90 | 0.088 | 0.030* |
| | AA | 1.85 | 6.50 | | 1.30 (0.37–5.08) | | | |
| | Allele | | | | | | | |
| | A | 32.58 | 47.10 | 2.17 (1.37-3.43) | – | 13.63 | 0.000* | 0.000* |
| | G | 67.42 | 52.90 | | – | | | |
| rs3780135 | Genotype | | | | | | | |
| | GG | 52.26 | 33.55 | | 1.00 | | | |
| | GA | 30.32 | 38.70 | 0.55 (0.39-0.95) | 1.19 (0.52–2.73) | 4.72 | 0.029* | 0.010* |
| | AA | 17.42 | 27.74 | | 0.97 (0.26–1.42) | | | |
| | Allele | | | | | | | |
| | G | 22 | 29 | 1.15 (0.72-1.83) | – | 4.11 | 0.043* | 0.014* |
| | A | 78 | 71 | | – | | | |
| rs12430881 | Genotype | | | | | | | |
| | AA | 65.16 | 61.93 | | 1.00 | | | |
| | GA | 25.80 | 18.06 | 2.52 (1.28 -4.95) | 1.09 (0.48–2.69) | 7.51 | 0.006* | 0.002* |
| | GG | 9.03 | 20 | | 1.03 (0.50–2.68) | | | |

**Notes.**
[a]Critical P value.
[b]Bonferroni corrected P value.
Significant values are shown in (*).

and contributes to the occurrence of disease. Our results are in accordance with (*Steinberg & Steinfeld, 1960*; *Urtishak et al., 2016*) which states that familial occurrence of leukemia exists (OR = 1.87, P = 0.050). Some studies found a correlation between parental exposure to radiation before conception, that may be due to their working environment (*Shu et al., 2002*). In our analysis, neither patients nor parents were ever exposed to radiations. Hepatomegaly and nephropathy are often seen in B-ALL subjects having chemotherapy. Malfunctioned leucocytes in the liver and kidney leads to enlargement of these organs (*Rasool et al., 2015*). Another study suggests that hepatomegaly and nephropathy may be the consequence of chemotherapeutic toxicity (*Giamanco et al., 2016*).

It is well established fact that cancer risk is influenced by numerous genetic variants having any risk or protective effect. The degree of penetrance of a certain genotype in the population and environmental factors is a major cause of cancer (*Fletcher & Houlston, 2010*). The information given by allelic and genotypic data of single nucleotide polymorphism in a population propose the possible genetic markers for cancer risk and predict possible targeted therapies (*Griffith et al., 2016*; *Wu & Li, 2018*).

**Table 5  Stratification analysis for association between genotypes and risk of B-ALL.** ORs were obtained from logistic regression model with adjustment for family history, smoking and consanguinity.

| | OR (95% CI) | | |
|---|---|---|---|
| **rs35958982** | **GG** | **GA** | **AA** |
| Family history status | | | |
|   Yes | 1 | 0.92(0.18–4.56) | 0.50(0.06–4.15) |
|   No | | | |
| Smoking status | | | |
|   Yes | 1 | 0.30(0.08–1.09) | 0.46(0.10–2.12) |
|   No | | | |
| Consanguinity status | | | |
|   Yes | 1 | 0.86(0.24–3.10) | 0.94(0.20–4.37) |
|   No | | | |
| **rs3780135** | **GG** | **GA** | **AA** |
| Family history status | | | |
|   Yes | 1 | 0.60(1.70–0.21) | 0.44(0.15–1.30) |
|   No | | | |
| Smoking status | | | |
|   Yes | 1 | 1.08(0.46–2.50) | 1.3(0.59–2.90) |
|   No | | | |
| Consanguinity status | | | |
|   Yes | 1 | 0.62(0.26–1.44) | 0.60(0.26–1.35) |
|   No | | | |
| **rs12430881** | **AA** | **AG** | **GG** |
| Family history status | | | |
|   Yes | 1 | 0.58(0.16–2.15) | 1.80(0.65–4.98) |
|   No | | | |
| Smoking status | | | |
|   Yes | 1 | 1.05(0.45–2.42) | 1.08(0.45–2.60) |
|   No | | | |
| Consanguinity status | | | |
|   Yes | 1 | 1.15(0.50–2.65) | 0.49(0.18–1.35) |
|   No | | | |

In this study, SNP rs35958982 is a germline polymorphism present in transmembrane region of *FLT3* gene. It is a non-synonymous variant which leads to the change in structure of the protein. High throughput DNA sequence analysis has been done to check the frequency of rs35958982 with leukemiogenesis in drivers and passengers which showed no association with AML (*Fröhling et al., 2007*). Present study in contrast displayed the association of SNP rs35958982 with the disease (OR = 2.30, CI [1.20–4.50], $P = 0.005$). Detailed analysis of genotype frequency in the population showed no association with B-ALL (OR = 3.67, CI [0.75–18.10], $P = 0.088$). This might be due to the fact that SNP rs35958982 is rare in acute lymphoblastic leukemia with low penetrance. Current study also depicts that individuals with risk allele A at locus 13q12.2 (rs12430881, *FLT3*) (OR = 1.15, CI [1.37–3.43], $P = 0.0426$) and genotype GG were more prone to B-ALL (OR =

2.52, CI [1.28–4.95], $P = 0.006$). It has been found that disruption of *FLT3* gene due to the presence of mutation or SNP leads to deficiency of B-lymphoid progenitors suggesting its critical role in survival and proliferation of blast cells (*Zriwil et al., 2018*).

*Bodian et al. (2014)* studied allele frequency of paired box domain (*PAX5*) polymorphism rs3780135 in different populations, i.e., African 34%, African European 49%, Central Asian 85%, East Asian 94%, European 95% and Hispanic 88%. Pakistan lies in South East Asia having frequency of rs3780135 (47.1%) which is lower than previously reported in East Asian population. *Firtina et al. (2012)* found polymorphism rs3780135 in B-ALL subjects with increased mRNA expression of *PAX5* suggesting the possible role of SNP with increased proliferation of blast cells. In Pakistani population, minor allele frequency was significantly identified in B-ALL subjects (OR = 2.17, CI [1.37–3.43], $P = 0.000$). Heterozygous genotype GA (38.7%) was more frequently identified in our cohort than homozygous risk genotype AA (27.74%) which manifested significant difference in frequency (CI [0.72–1.83], $P = 0.029$) and also showed protective effect (OR = 0.55). *PAX5* is involved in repression of T-cells, activation of B-cell proliferation from blast cell therefore, presence of any variant in this gene affects its pathway which may leads to increased expression of *PAX5* and results into B-ALL (*Firtina et al., 2012*).

## CONCLUSIONS

The findings of the present study significantly demonstrate that SNPs rs35958982 and rs12430881 correlate with the increase risk of B-ALL, however, SNP rs3780135 has a protective effect. The environmental risk factors of B-ALL, including family history, parental consanguinity and smoking, are found to have an imperative role in progression of disease. Although the data is balanced, but not robust, the small cohort of subjects limits the conclusion of article. To eliminate this limitation and validate the results of present study, larger prospective studies need to be conducted in the same ethnic group. Furthermore, various other demographic and environmental factors should also be considered and appraised for their association with B-ALL.

## ACKNOWLEDGEMENTS

Authors would like to acknowledge The Children's Hospital and Institute of Child Health, Lahore, Pakistan for their assistance and support during blood sample collection.

### Funding
The authors received no funding for this work.

### Competing Interests
The authors declare there are no competing interests.

## Author Contributions

- Ammara Khalid conceived and designed the experiments, performed the experiments, analyzed the data, contributed reagents/materials/analysis tools, prepared figures and/or tables, authored or reviewed drafts of the paper, approved the final draft.
- Sara Aslam performed the experiments, prepared figures and/or tables.
- Mehboob Ahmed analyzed the data, contributed reagents/materials/analysis tools, authored or reviewed drafts of the paper.
- Shahida Hasnain contributed reagents/materials/analysis tools, authored or reviewed drafts of the paper.
- Aimen Aslam analyzed the data, prepared figures and/or tables.

## Ethics

The following information was supplied relating to ethical approvals (i.e., approving body and any reference numbers):

The University of the Punjab, Lahore, Pakistan granted Ethical approval to carry out the study within its facilities (Ethical Application Ref: sbs/222/18).

## Data Availability

Data is available as Supplemental File.

## Supplemental Information

Supplemental information for this article can be found online at http://dx.doi.org/10.7717/peerj.7195#supplemental-information.

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
