# Peer review of "Risk assessment of FLT3 and PAX5 variants in B-acute lymphoblastic leukemia: a case–control study in a Pakistani cohort"

_PeerJ, doi:10.7717/peerj.7195_

## Round 0.1 · original submission · Major Revisions

Please take the help of a native speaker or editing services to improve the overall MS readability and also get the statistician to check your data and include his suggestions and inputs on multiple testing and any other relevant statistics in the manuscript and update the literature.

Reviewer 1 ·

Basic reporting

1. The article is not written in clear English. It contains many mistakes. Sentences are partly wrong in grammar.
2. references are not up to date. Citations are either old or not appropriate. Some important studies on ALL are missing at all. Please, go and check pubmed.
3. The format is acceptable. However, the authors mention the education of the parents was taken into account with reference to Table 1, but there is no such information.
4. The authors give statements on the prevalence of SNP's genotypes/alleles. This is clearly wrong. If at all, it should be "frequencies".
5. A part of the introduction (lines 43-55) seems to be simply copied as the language is completely different from the rest of the manuscript.

Experimental design

The aims are clear, but the statistical analysis was not performed on a high standard.
Education of the parents is missing in the results, even though mentioned in the manuscript. The material could have been analysed with a multivariate model with covariates included.
Besides the methods are described in a bad way.

Validity of the findings

The data are not analysed in an appropriate way. There should be a more general table showing the material used. The table 3 with the results is not clear to me. Many numbers are missing. For instance, 16.77 % have a positive family history, 52.9 have a negative family history. What about the remaining 30%?
Thus, the conclusions are not sufficient and clear enough.

Additional comments

First of all, the idea of the manuscript is fine, given the overall importance of B-ALL in Pakistan.
However, the statistics should be approved by a statistician. Multiple testing has not been considered (3 SNPs). The testing should have been performed on a multivariate level with alleles/genotypes and other covariates included. More recent literature should have been added.

·

Basic reporting

The authors evaluated the association between DNA single nucleotide polymorphisms (SNPs) in PA5X and FLT3 genes with B-ALL in a case control study with equal number of cases and controls. The study is appropriately designed but the article is poorly written and the results lack clarity.
1- The whole article needs a thorough proofreading as it contains many spelling errors /misspelled words and grammatical mistakes that are at times changing the context. Few examples:
I. Title: line1 says coharts” instead of cohorts
Risk assessment of FLT3 and PAX5 variants in B-acute lymphoblastic leukemia: A case-control study in a Pakistani cohart
II. Abstract
Prevalence, association and frequency have been used interchangeably which gives the impression the authors don’t know what they are talking about.

Abstract RESULTS: Risk allele frequency ‘A’ of FLT3 missence (missense??) polymorphism rs35958982 showed significantly more prevalent (should be prevalence) in B-ALL subjects (p=0.00526)
proginator

III. Introduction (words in parentheses)misspelled or grammatically incorrect)
Line 56=>In hematologic (melangices), 70% to 100% increased expression of FLT3 in acute (myloid) leukemia (AML) and acute lymphoblastic leukemia (ALL) are reported previously
These studies signify the critical role of FLT3 expression in (proliferaltion) of blast cells (Carow et al., 1996).
B-cell-specific activator protein (PAX5) encode transcription factors that are the member of paired box domain. PAX5 plays imperative role in the (commintment) of B-cell lineage from blast cells as it controls the differentiation

(proginator cells) has been used several times instead of progenitor cells
IV. “Figure 1 -3”

DNA ladder has continuously been referred to as “Leader”
(Moreover figure 2 is not properly labelled)
And countless other examples like this are present within the body of the manuscript showing that the article has not been proofread.

Experimental design

2- Materials and Methods:

I. Discrepancy in the number of patients between abstract and M&M
155 patients and 155 controls =310 patients not 300.

II. What was the criterion used for the selection of SNPs for the study?
III. Was the association analysis adjusted for the family history, smoking and Consanguinity?
IV. Descriptive stats with respect to age sex and family history should be clearly explained in the form of a table

Validity of the findings

3- Results
Table 4 is confusing; one cannot comprehend whether the ORs presented are for allelic or genotypic model. 95% CI in the case of rs35958982 is very wide (For OR=3.67, CI= 0.75-18.10) what could be the cause of it? Was it inadequate sample size or Software failure?

4- Discussion
I. Ultrasounds are not radiations
II. In general a more detailed discussion of the results would be valuable including the presentation of hypotheses linking the SNPs identified in the study and increased risk of B-ALL. The biological plausibility leading from the selected SNPs to the ALL should be stressed.

---

## Round 0.2 · Minor Revisions

Both the reviewers felt that their comments on the language and grammar was not adequately addressed. Please also tone down your conclusions as this rather a small set and needs further confirmation in a validation cohort.

Please answer all the points raised by the reviewer 1 point by point and submit with your explanations/analysis/answers.

Reviewer 1 ·

Basic reporting

The comments of the first review have been taken into account. Literature has been updated. Analysis have been improved. However:

1. Despite the comments in the rebuttal letter, the English has not been improved very much. There are still many mistakes in grammar (i.e. there are far too many commas) like in line 185: "According to previous studies, association of First and second-degree..."
This should be: "According to previous studies association of first and second-degree..."
in line 106: "subjects greater than 15 years" should be "subjects older than 15 years"
in line 26: "PAX5 polymorphism rs3780135 risk allele ‘A’ frequency was more in B-ALL subjects than ancestral allele frequency ‘G’". This should be rewritten.
2. There is no consistency about gene names (FLT3 and PAX5 are either normal or italic), while they should be italic through the entire manuscript.
3. There is no consistency about blanks (before and after commas, sometimes blanks are missing at all, sometimes there too many blanks.) This also refers to the tables, in which the numbers should not simply be centered, but the dots of digital numbers should be in line.
4. There is no consistency about the numbers (i.e. Table 3: 0.6(.26-1.35) or 0.497(0.18-1.35). It should be: 0.60 (0.26-1.35) with two digits after the dots.
5. Overall, the readability could be improved by adding some articles (the and a, i.e. in the present study....)
6. The following sentence attached to Table 1 should be rewritten: "Parameters with P<0.05 is significant studied are linked to B-ALL. (*) show significant."

7. Structure: the entire information about genotyping (lines 121 - 145) should be shown as a table.
8. Remarks about education: You state: "Our result showed that mothers (39.4%) and fathers (16.3%) were educated." As this information is not used in further analysis, I would take this statement out.

Experimental design

Methods:
Overall, one must admit, that the number of cases and controls is rather small, which will limit the conclusion of the manuscript. This also holds for other risk factors such as family history, smoking or parental consanguinity. If the numbers would be ten-fold, these risk factors would be for sure significant like it has been shown in other publications before.

1. The idea to perform a multivariate logistic regression model has been applied. However, the intention was not to check for interactions. This is not relevant for material of this size. The clear intention was to check, whether SNPs would still show significant effects if corrected for environmental effects. Thus, you should leave out the interaction topic.
2. As the authors deal with several SNPs, there should be a correction applied for multiple testing. Please, check the Bonferroni corrected P-values! p*<p/n, where n is the number of tests performed.

Validity of the findings

Overall, one must admit, that the number of cases and controls is rather small, which will limit the conclusion of the manuscript. Thus, the authors should be careful with their conclusions and statements. The data is not robust, even though it is balanced.
Although I know that it might be difficult to collect a sample set, I would suggest to validate these results in a second data set. One could also split the original data into two random sets. However, as the set has already been analysed, the results would be biased.

Authors mentioned

Table 3: Family history: OR and CI are not correct: 0.6(1.7-0.21). As stated above, please check for consistency. (.26 vs. 0.26).
Take out the P (interaction).

Additional comments

The authors have presented a small overview on the risk assessment of FLT3 and PAX5 variants in B-acute lymphoblastic leukemia. Their case - control study has been performed in a Pakistani cohort. Due to the fact, that the material is small, conclusions are limited for both factors - for the genetic factors such as the SNPs as well as for the environmental factors.
One important overall comment: authors should avoid the following statements like: "that FLT3 SNP rs35958982..." SNPs might be close to a gene, SNPs might be even in the identical genomic region, but SNPs are not identical with a gene as such. one suggestion: for locus 3p22.1 (rs9848754, ULK4) P-values reached significance.

·

Basic reporting

The authors have made satisfactory improvements to the manuscript and the results are more clearly presented and most errors have been taken care of.

Experimental design

Satisfactory

Validity of the findings

satisfactory

Additional comments

Please refer to the annotated pdf

---

## Round 0.3 · accepted · Accept

As all the queries raised by the reviewers and upon internal evaluation it was felt that the manuscript is now suitable for publication in PeerJ